# Pontus: A Memory-Efficient and High-Accuracy Approach for Persistence-Based Item Lookup in High-Velocity Data Streams

## ABSTRACT

In today's web-scale, data-driven environments, real-time detection of persistent items that consistently recur over time is essential for maintaining system integrity, reliability, and security. Persistent items often signal critical anomalies, such as stealthy DDoS and botnet attacks in web infrastructures. Although various methods exist for identifying such items as well as for determining their frequency, they require recording every item for processing, which is impractical at very high data rates achieved by modern data streams. In this paper, we introduce Pontus, a novel approach that uses an approximate data structure (*sketch*) specifically designed for the efficient and accurate detection of persistent items. Our method not only achieves fast and precise lookup but is also flexible, allowing for minor modifications to accommodate other types of persistence-based item detection tasks, such as detecting persistent items with low frequency. We rigorously validate our approach through formal methods, offering detailed proofs of time/space complexity and error bounds to demonstrate its theoretical soundness. Our extensive trace-driven evaluations across various persistence-based tasks further demonstrate Pontus's effectiveness in significantly improving detection accuracy and enhancing processing speed compared to existing approaches. We implement Pontus in an experimental platform with industry-grade Intel Tofino switches and demonstrate the practical feasibility of our approach in a real-world memory-constrained environment.

## KEYWORDS

Data stream processing, persistent item lookup, probabilistic data structure

## ACM Reference Format:

Anonymous Author(s). 2024. Pontus: A Memory-Efficient and High-Accuracy Approach for Persistence-Based Item Lookup in High-Velocity Data Streams. In *Proceedings of ACM Conference (WebConf'25)*. ACM, New York, NY, USA, 13 pages. https://doi.org/10.1145/nnnnnnn.nnnnnnn

## 1 INTRODUCTION

Recognizing persistent items is vital across a range of web-scale applications, such as analyzing user behavior in e-commerce platforms [11], preventing fraud in online financial transactions [4], and detecting anomalies in network traffic [19, 23]. For example, certain web-based network threats utilize stealthy techniques by distributing malicious packets at a controlled rate over an extended period,

rather than overwhelming targets in a short burst. This tactic is specifically designed to evade detection by traditional web security monitoring methods [25]. In a different context, persistent items in server log files or monitoring data can reveal recurring faults or performance bottlenecks in distributed web systems [34]. Identifying and analyzing these persistent anomalies allows engineers to address systemic issues that may not surface during standard testing, but which consistently degrade the performance of web services. Similarly, persistent user actions on web applications—such as frequently used features or repeatedly accessed content—can inform developers in optimizing UI/UX designs [20, 48], enhancing user satisfaction and engagement on a broad scale.

Beyond detecting persistent items, it is also important to identify variations, such as items that are both persistent and frequent [12, 31], or persistent and infrequent [18]. For example, persistent and frequent requests in web services may indicate critical resources or endpoints. Identifying these patterns can help optimize load balancing and server provisioning to handle peak traffic more effectively [21]. In enterprise networks, web services like Fast Reverse Proxy (FRP) [37] expose local servers behind firewalls [6] to external traffic. FRP connections, while consistent, often generate low packet volumes, making them persistent yet infrequent [18], which can be key to enhancing security and network performance.

**Challenges:** Detecting persistence-based items in real-time poses significant challenges due to three primary factors. (i) The sheer volume and high speed of data streams make it impractical to track all items for analysis and lookup [45]. (ii) Many modern computing systems used for web-scale data mining have tight memory budgets and require highly compact processing methods. These systems rely on on-chip CPU caches for high-speed processing, but are constrained by the limited capacity of the fastest L1 cache [31]. Similarly, programmable network hardware that is posed to permeate future-generation communication infrastructures is extremely resource-constrained [3, 38]. (iii) The item distribution is typically highly skewed [50], meaning that most items are non-persistent and only a small portion is persistent.

**Limitations of Prior Art:** To overcome these obstacles, approximate data structures, commonly referred to as *sketches*, have been adapted to data stream processing in high-speed environments [13, 16, 26, 28, 39–41, 44, 46, 49]. Several sketch-based schemes target persistence-based item lookup [12, 18, 29–32, 48]. However, as expounded in Section 7, these methods face the following key limitations. (i) Many employ coarse item replacement strategies, such as the direct swap strategy [48], which often lead to persistent items being prematurely evicted from buckets due to frequent collisions with non-persistent items, especially under tight memory constraints where hash collisions are more severe. (ii) Some methods track multiple features per item to better protect persistent items, but this compromises memory efficiency [30–32]. (iii) Other approaches involve complex update operations, such as matrix

multiplication, reducing processing speed and limiting their ability to handle high-speed data streams [15]. (iv) Most methods are overly dependent on multiple tuning parameters, making them less practical for real-world deployment [12].

**Our Solution:** In this paper, we introduce Pontus, a novel probabilistic method designed for the accurate and efficient lookup of persistent items. Pontus introduces several key novelties in sketch design for persistence-based item lookup: (i) it employs new flags to avoid double-counting items and excessive decay of persistence values due to severe hash collisions; (ii) it implements a probability-decay eviction strategy prioritizing the removal of non-persistent items that tend to dominate real-world data distributions [7, 46, 50]; (iii) it handles persistent item lookup tasks in scenarios with long item keys by introducing a fingerprint-based variant that utilizes a counter merge technique to enhance memory efficiency [45].

We rigorously model and analyze Pontus, offering theoretical guarantees on error bounds, time complexity, and space complexity, enabling us to quantify its performance and reliability.

We also implement a prototype of Pontus in C++ and evaluate its performance across diverse datasets and tasks, including persistent item lookup, persistence estimation, and the detection of both persistent and frequent, as well as persistent and infrequent, items. Our results demonstrate Pontus's robustness and superiority over previous sketch-based designs. For persistent item lookup, Pontus attains the highest detection accuracy even compared with the leading competitors, P-Sketch [30] and Stable-Sketch [32]. In persistence estimation, it reduces estimation errors by up to 484.2% relative to existing methods. When detecting persistent and (in)frequent items, Pontus achieves the best F1 score of approximately 0.9, even with a constrained memory budget of 64KB. Moreover, thanks to its streamlined update process, Pontus delivers the fastest update speeds, outperforming benchmark schemes across various tasks.

Finally, we deploy Pontus on a hardware Tofino switch using the P4 language [8], where it achieves a recall of 0.95 for persistent item lookup with 16,384 entries by consuming only 8.2% of the total available switch resources on average and entailing an average packet processing latency of 409ns. This demonstrates Pontus's applicability to practical resource-constrained systems.

## 2 PROBLEM DEFINITION

**Data Stream:** We consider a data stream $\mathcal{U} = \{e_1, e_2, \ldots, e_n\}$ composed of various items, where each item is represented as a key-value pair. The key serves as the item identifier, and value is the corresponding value associated with that item. For instance, in network monitoring, the key often represents a flow identifier, such as a source-destination address pair [42], while the value could be its count information, such as frequency or persistence.

**Sketch:** Sketches are probabilistic summary data structures that track values in a fixed number of entries known as *buckets*. Classic examples of sketches include the Count-Min Sketch [14], Count Sketch [9], and CU Sketch [17], among others. A Count-Min Sketch is represented by a two-dimensional array of buckets with $w$ columns and $d$ rows. Initially, each counter in the buckets array is set to zero. Additionally, $d$ pairwise-independent hash functions $h_1, \ldots, h_d$ are chosen, with each hash function corresponding to a row. When an item $(e_i, v_i)$ arrives, indicating that item $e_i$ is updated by a quantity of $v_i$, $v_i$ is added to the counter in a bucket

in each row; the specific bucket is determined by applying the corresponding hash function $h_z$ to $e_i$. Formally, $\forall z, 1 \leq z \leq d$, $B(z, h_z(e_i)) \leftarrow B(z, h_z(e_i)) + v_i$. During the query process, the count of item $e_i$ is estimated as $\hat{S}_{e_i} = \min_z [B(z, h_z(e_i))]$.

**Persistence:** In data streams, persistence is defined across $M$ non-overlapping, consecutive time windows. The window size can be adapted based on data volume or timestamps. For example, in network traffic analysis, one can define each window by a fixed number of packets, such as $N = 1000$ packets per window, meaning that every 1000 received packets constitute one window, while the data volume in each window can vary. The persistence $P_{e_i}$ of an item $e_i$ increments by 1 if $e_i$ appears at least once in a window, regardless of how many times it occurs within that window.

**Persistent Item Lookup:** An item $e_i$ is said to be $\alpha$-persistent if its persistence $P_{e_i}$ satisfies $P_{e_i} \geq \alpha M$, where $\alpha \in (0, 1]$ is a user-defined threshold. This means that $e_i$ must appear in at least a fraction $\alpha$ of the total number of time windows $M$ by the measurement point to be classified as persistent.

**Persistence Estimation:** The task of persistence estimation involves accurately determining the value of $P_{e_i}$ for each unique item $e_i$ in the data stream $\mathcal{U}$.

**Persistent and Frequent Item Detection:** For each $e_i \in \mathcal{U}$, let $S_{e_i}$ denote the total number of occurrences (frequency) of $e_i$ in the data stream. Given a frequency threshold $\beta \in (0, 1]$, an item $e_i$ is considered $\beta$-frequent if its frequency $S_{e_i}$ satisfies $S_{e_i} \geq \beta S$, where $S = \sum_{e_i \in \mathcal{U}} S_{e_i}$ is the total number of occurrences (frequency sum) across all items in the stream. The set of items that are both frequent and persistent is then $\mathcal{F}_\alpha^\beta = \{e_i \in \mathcal{U} \mid S_{e_i} \geq \beta S \text{ and } P_{e_i} \geq \alpha M\}$.

**Persistent and Infrequent Item Detection:** Given an additional frequency threshold $\kappa \in (0, 1)$, considerably lower than $\beta$, we define the set of persistent but infrequent items as follows: $\mathcal{I}_\alpha^\kappa = \{e_i \in \mathcal{U} | S_{e_i} < \kappa S \text{ and } P_{e_i} \geq \alpha M\}$.

## 3 DESIGN

### 3.1 Principles

State-of-the-art methods for persistent item lookup, such as P-Sketch [30], Stable-Sketch [32], and Tight-Sketch [31], use a dual-dimensional feature approach to better protect potential persistent items, which is detailed in Section 7. While this improves detection accuracy, it reduces memory efficiency, resulting in fewer available buckets. This problem is exacerbated in memory-constrained environments, where hash collisions become more frequent.

Pontus improves the state of the art above by enhancing both memory efficiency and detection accuracy. Given the skewed nature of real-world data distributions [32], we employ a probability-decay eviction strategy to prioritize the retention of potentially persistent items. Our design incorporates two flags (requiring only 2 bits) and is based on three key principles: (i) Since the persistence of an item increases by only 1 per time window, we introduce an ***arrival flag*** $F$ to track whether the recorded item has arrived during the current window. (ii) Unlike item frequency, which can grow rapidly, item persistence increases more slowly, making persistent items more vulnerable to eviction in cases of frequent hash collisions from a large number of incoming items. To prevent this, we limit bucket access to allow persistence decay only once per window. The ***collision decay flag*** $R$ indicates whether a hash collision

**Figure 1: Data structure of Pontus.**

has occurred in the bucket during the current window. If a decay has already occurred, further decay from other items is ignored to avoid excessive reduction in the tracked item's persistence. (iii) If the persistence of a tracked item has been reduced due to a hash collision, its persistence is increased by 2, rather than 1, when the item arrives in the same window. Notably, our method requires only one additional flag $R$ compared to existing approaches [48], yet it delivers superior detection performance, as demonstrated in Section 5. Additionally, Pontus's efficient design facilitates its practical deployment on resource-constrained hardware, as proven in Section 6 with a P4-programmable Tofino switch.

### 3.2 Data Structure

Here, we illustrate the data structure of Pontus using persistent item lookup as an example. We will elucidate how our scheme can be extended to handle other persistence-based tasks in Section 3.5.

As shown in Figure 1, our probabilistic data structure consists of $d$ rows and $w$ columns. Each bucket, denoted as $B(i, j)$ for $1 \le i \le d$ and $1 \le j \le w$, contains four fields: $K_{i,j}$ stores the key of the current candidate persistent item hashed to the bucket, $P_{i,j}$ tracks the persistence count of the item, $F_{i,j}$ is *the arrival flag*, and $R_{i,j}$ represents *the collision decay flag*. Since each flag occupies only one bit, both flags can be packed into a single 8-bit machine word for efficient deployment. Our method utilizes $d$ pairwise-independent hash functions, denoted as $h_1, \ldots, h_d$, where each $h_i$ ($1 \le i \le d$) maps the key of each incoming item to one of the $w$ buckets in the $i$-th row. Notably, the size of this data structure is fixed and can be pre-allocated in memory for efficient operation.

### 3.3 Basic Operation

Pontus hinges upon two fundamental operations: (i) Update, involving the insertion of each incoming item into the probabilistic data structure; and (ii) Query, enabling the retrieval of the estimated persistence of a given item.

*3.3.1 Update.* Algorithm 1 outlines the update process of Pontus for each time window. At the beginning of each window, the status of all flags in each bucket is reset to $True$ (Line 1). Subsequently, three scenarios may occur:

(i) Upon the arrival of an item $e_i$, the algorithm uses the hash functions $h_1, h_2, \ldots, h_d$ iteratively to locate an appropriate bucket $B(i, j)$. If the hashed bucket is empty, denoted by $K_{i,\text{index}} == NULL$, item $e_i$ is inserted into the bucket, and its persistence counter $P_{i,\text{index}}$ is initialized to 1. The flag $F_{i,\text{index}}$ of the bucket is set to $False$ to indicate that the item has been added in current time slot. Similarly, the flag $R_{i,\text{index}}$ is set to $False$, indicating that the bucket cannot be decayed by other items within the current time window (Lines 2-5). Then the hash operations cease (Line 6). Unlike methods such as [14, 42], which track an incoming item across all rows, we track

---

**Algorithm 1:** Update Process in Each Time Window

---

**Input:** an item $e_i$, hash functions $h_1, h_2, \ldots, h_d$, $\min_p \leftarrow +\infty$

1 **Initialization (performed once before the start):** Initialize the persistence counters $P_{i,j}$ to 0, item keys $K_{i,j}$ to NULL, and flags $F_{i,j}, R_{i,j}$ to True for all buckets $B(i, j)$.

2 **for** $i = 1$ *to* $d$ **do**

3      index $\leftarrow h_i(e_i.\text{key})$

     // Case 1: Empty bucket

4      **if** $K_{i,index} == NULL$ **then**

5          $K_{i,index}, P_{i,\text{index}}, F_{i,\text{index}}, R_{i,\text{index}} \leftarrow e_i.\text{key}, 1, False, False$

6          **return**

     // Case 2: Item already tracked in the bucket

7      **else if** $K_{i,index} == e_i.key$ *and* $F_{i,index} == True$ **then**

         // Persistence decayed by another item

8          **if** $R_{i,index} == False$ **then**

9              $P_{i,\text{index}} \leftarrow P_{i,\text{index}} + 2$ // Reduced

10          **else**

11              $P_{i,\text{index}} \leftarrow P_{i,\text{index}} + 1$ // Not reduced

12          $F_{i,\text{index}}, R_{i,\text{index}} \leftarrow False, False$

13          **return**

     // Track bucket with minimum persistence

14      **else if** $P_{i,index} < \min_p$ **then**

15          $\min_p$, minRow, minIndex $\leftarrow P_{i,\text{index}}, i,$ index

     // Case 3: Probabilistic replacement

16 **if** $R_{minRow,minIndex} == True$ **then**

17      **if** $random(0, 1) < \frac{1}{P_{minRow,minIndex}+1}$ **then**

18          $P_{\text{minRow,minIndex}} \leftarrow P_{\text{minRow,minIndex}} - 1$

19          $R_{\text{minRow,minIndex}} \leftarrow False$

20          **if** $P_{minRow,minIndex} == 0$ **then**

21              $K_{\text{minRow,minIndex}} \leftarrow e_i.\text{key}$

22              $P_{\text{minRow,minIndex}} \leftarrow 1$

23              $F_{\text{minRow,minIndex}} \leftarrow False$

24 **return**

---

each item in only one row. This optimizes memory usage by freeing up more buckets to track other items.

(ii) If the hashed bucket contains $e_i$ and its corresponding flag $F_{i,\text{index}}$ is $True$, indicating that the item has not arrived within the current time window, the increase of the persistence counter $P_{i,\text{index}}$ follows two scenarios: (1) If $e_i$ arrives late in the current time window and its counter has been reduced by another item that hashed into the same bucket (as indicated by $R_{i,\text{index}}$ being $False$), the persistence counter for $e_i$ will be increased by 2 to compensate for underestimation errors (Lines 7-9). (2) Otherwise, the counter will be increased by 1 (Lines 10-11). Then the flag $F_{i,\text{index}}$ is updated to $False$ to signify that the item has arrived, and $R_{i,\text{index}}$ is set to $False$, signaling that the bucket is inaccessible to other items within this time window (Line 12). The hash operations halt (Line 13).

(iii) In cases of hash collisions across all rows, our method selects the bucket with the smallest persistence counter to resolve the collision (Lines 14-15). Probabilistic decay is triggered when a randomly generated value is less than the reciprocal of the persistence counter plus one. If the persistence counter successfully decrements by 1, the flag $R_{\text{minRow,minIndex}}$ is set to $False$ (Lines 16-19), ensuring that other items hashing into this bucket cannot

perform additional decay operations. If the persistence counter reaches zero, the bucket is updated with the new item: the key is replaced, the persistence counter is reset to 1, and the flag is set to *False* (Lines 20-23). If the counter does not reach zero, the new item is discarded. This approach favors items with higher persistence, reducing their chances of being replaced.

*3.3.2 Query.* To identify persistent items, the algorithm checks the condition $\exists i \in \{1, 2, \ldots, d\}, \exists j \in \{1, 2, \ldots, w\}$ such that $P_{i,j} \geq \alpha M$ for each item $e_i$. If this condition is met for any item $e_i$, then that item is identified and reported as a persistent item.

## 3.4 Running Examples

To exemplify the update process, we provide running examples in Figure 2, using a sketch with two rows and three columns each.

❶ When item $e_1$ arrives, it uses the hash function $h_1$ to find a bucket in the first row. Since the bucket is empty, $e_1$ is inserted, and hashing stops. The bucket's status updates to $(e_1, 1, F, F)$, indicating no further increase in the persistence of $e_1$ during the current time window, and preventing other items hashed into this bucket from performing probabilistic decay operations.

❷ Item $e_4$ arrives and also utilizes $h_1$ to locate a bucket in the first row. It finds a match, ending the hash operations. A check shows that the flag $F$ is *False*, indicating that $e_4$ has already been logged in the current time window, thus $e_4$'s status remains unchanged.

❸ Item $e_6$ arrives and uses hash function $h_2$ to find a matching bucket. The flag $F$ is *True*, indicating $e_6$ has not arrived in the current time window. However, the flag $R$ is *False*, i.e., another item has hashed into this bucket and decayed the persistence counter. Upon $e_6$'s arrival, its persistence counter increases by 2 to ensure accuracy. The bucket's status updates from $(e_6, 3, T, F)$ to $(e_6, 5, F, F)$.

❹ When item $e_8$ arrives, it uses hash functions $h_1$ and $h_2$ iteratively to locate an available bucket, yet both buckets are occupied. Since $e_7$ has lower persistence and its $R$ flag is set to *True* (i.e., the bucket has not been decayed by other items in the current time window), $e_8$ attempts to decay its counter. The decay probability is $\frac{1}{4+1}$. If the decay is unsuccessful, the bucket status remains unchanged. If successful, it changes from $(e_7, 4, T, T)$ to $(e_7, 3, T, F)$. If $e_7$'s counter decays to 0, $e_8$ can evict and replace $e_7$ in the bucket.

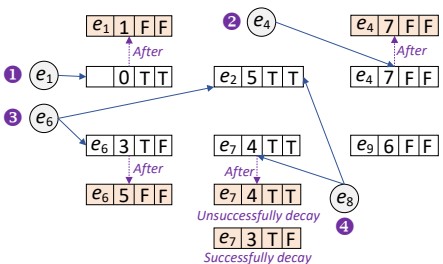

**Figure 2: Running examples of Pontus's update process.**

## 3.5 Adaptation to Other Lookup Tasks

*3.5.1 Persistence Estimation.* For persistence estimation, we employ the same update procedure as used in persistent item lookup. During the query process, we hash all items in the data stream. If a match is found in the sketch, we report the corresponding value as the estimated persistence. However, if no match is found, the smallest value among the hashed buckets in all rows is chosen to mitigate overestimation errors.

*3.5.2 Persistent and Frequent Item Lookup.* To achieve lookups that combine persistence and frequency, we introduce an additional counter $V_{i,j}$ to each bucket to track the frequency of each recorded item. When a new item arrives, the update process for persistence follows the same steps as described in Section 3.3. The item first attempts to locate an empty or matching bucket. If successful, the item's frequency is incremented by 1. In the case of hash collisions across all rows, the item seeks to decrement the frequency and persistence values of the bucket with the smallest sum of frequency and persistence, guided by the probability $1/(V_{i,j} + P_{i,j} + 1)$. This probability ensures that items with high persistence and frequency are more likely to remain in their respective buckets. Since frequency is always equal to or greater than persistence, if the persistence value reaches 0, the newly arrived item replaces the tracked item, resetting both the frequency and persistence counters to 1. The query process is similar to that of persistent item lookup, requiring a scan of all buckets to identify items where both the persistence and frequency exceed their respective thresholds.

*3.5.3 Persistent and Infrequent Item Lookup.* To formalize this task, we introduce an objective function $\Omega_{i,j} = P_{i,j} - \varsigma V_{i,j}$, where $\varsigma$ represents a weight parameter (e.g., 0.2). In this formulation, a potential persistent item with higher frequency receives a lower reward. Consequently, persistent and infrequent items tend to have larger objective values. During the update process, if a bucket is successfully located, the procedure remains the same as described previously. In the event of hash collisions across all rows, the item selects the bucket with the smallest $\Omega$ value and decreases both the persistence and frequency counters by 1, based on the probability $1/(\Omega + 1)$. When the persistence counter is reduced to 0, the newly arrived item replaces the tracked item, and both the frequency and persistence counters are set to 1. The query process involves examining all buckets to identify items with persistence exceeding the persistence threshold and frequency below the frequency threshold.

## 3.6 A Variant for Higher Memory Efficiency

In scenarios where item keys are long and fast memory is limited, we propose a memory-efficient variant of Pontus, as follows.

*3.6.1 Fingerprint-Based Key Compression.* To ensure good invertibility, which means returning all persistence-based items solely from the sketch data structure without redundant hash operations, Pontus selects tracking the item key by default. However, this strategy may result in increased memory consumption, especially in scenarios with longer keys like 5-tuples in network data. To mitigate this issue, the proposed variant employs a hash function to generate a concise sequence of bits from the key, known as the fingerprint [45], which conserves memory and thus increases the number of available buckets for item storage. The utilization of fingerprints, however, introduces the risk of fingerprint collisions, where multiple items share the same fingerprint.

Formally, let $n$ be the number of items and $w$ the number of buckets in a row in the sketch data structure. Each bucket is equipped with an $\vartheta$-bit fingerprint generated by a hash function applied to the item keys. The probability of a fingerprint collision can be expressed as follows: $Pr\{fingerprint\ collision\} = 1 - (1 - 2^{-\vartheta})^{\frac{n}{w}}$. Table 1 shows the fingerprint collision rates for various numbers of items and fingerprint lengths. Each row corresponds to a specific number of buckets, $w = 750$, which reflects the configuration of

**Table 1: Collision rate with varying fingerprint lengths.**

| $n$ \ $\vartheta$ (bits) | 4 | 8 | 16 | 20 | 25 | 32 |
|---|---|---|---|---|---|---|
| 0.5M | 0.4748 | 0.0905 | $2.2 \times 10^{-3}$ | $4.9 \times 10^{-4}$ | $5.7 \times 10^{-5}$ | $3.7 \times 10^{-8}$ |
| 1M | 0.6643 | 0.1667 | $4.4 \times 10^{-3}$ | $9.8 \times 10^{-4}$ | $1.2 \times 10^{-4}$ | $3.7 \times 10^{-8}$ |
| 2M | 0.8592 | 0.2946 | $8.8 \times 10^{-3}$ | $2.0 \times 10^{-3}$ | $2.3 \times 10^{-4}$ | $3.7 \times 10^{-8}$ |
| 3M | 0.9459 | 0.4117 | $1.3 \times 10^{-2}$ | $2.9 \times 10^{-3}$ | $3.4 \times 10^{-4}$ | $1.4 \times 10^{-6}$ |
| 4M | 0.9777 | 0.5153 | $1.8 \times 10^{-2}$ | $3.9 \times 10^{-3}$ | $4.6 \times 10^{-4}$ | $3.0 \times 10^{-6}$ |

Pontus under a tight memory constraint of 16KB. As the fingerprint length $\vartheta$ increases, the probability of collision decreases rapidly. When the fingerprint length reaches 20 bits, the collision probability becomes negligible even with larger item counts $n$.

*3.6.2 Field Consolidation.* We take persistent item lookup as an example. In the default version of Pontus, we construct each field separately. However, this scheme can lead to memory under-utilization on widely-used x86 systems, where the memory allocation of each field is aligned to the machine word size, such as 8-bit, 16-bit, or 32-bit. For example, suppose we need to store five fields: a 19-bit fingerprint ($\varphi_1 = 19$), two 1-bit flags ($\varphi_{2-4} = 2$), and a persistence value up to 2000 ($\varphi_5 = 11$ bits). The total space required would be $\sum_{x=1}^{5} \varphi_x = 32$ bits. Yet, if we allocate each field separately, the memory usage would be much higher - 32 bits for the fingerprint, 8 bits for each flag, and 16 bits for the persistence value.

To improve memory utilization, we can consolidate these fields into a single machine word, provided that the total number of bits required for all fields does not exceed the word size, typically 32 or 64 bits. Formally, let $\varphi_x$ be the length (in bits) of the $i$-th field, where $x = 1, 2, \ldots, \zeta$, and $\zeta$ is the total number of fields. If the following condition is satisfied: $\sum_{x=1}^{\zeta} \varphi_\zeta \leq \Phi$, where $\Phi$ is the machine word size, then we can proceed to consolidate the fields into a single unit. By adopting this approach whenever possible, we can effectively utilize the available memory resources and avoid wastage caused by the fixed-size memory allocation for individual fields.

*3.6.3 Trade-off.* Although this variant enhances memory efficiency and improves lookup accuracy under limited memory conditions, the incorporation of fingerprint-based compression and counter consolidation operations slow down processing speed, an essential metric in data stream mining. This creates a trade-off: users prioritizing processing speed may prefer the basic method, while those valuing accuracy over speed may adopt the variant. We will delve into this trade-off in detail in the Section 5.4.

## 4 THEORETICAL ANALYSIS

We present a comprehensive theoretical analysis of Pontus, focusing on its application to persistent item lookup. This includes formal proofs of performance guarantees—covering space, update, and query complexity—and error bounds for both the default and memory-efficient variants. Empirical tests further validate these error bounds. Due to space limitations, only the error bounds are provided here, with additional results and detailed proofs included in Appendix A.

### 4.1 Error Bound of Pontus

THEOREM 4.1. *Let $e_i$ denote the item with the $i$-th highest persistence among all considered persistent items. Note that our analysis of the error bound specifically focuses on those persistent items. Given a small positive number $\epsilon$ and a persistent item $e_i$, we define the*

probability that the difference between $P_{e_i}$ and $\hat{P}_{e_i}$ exceeds $\epsilon P$ by the following inequality. Here, $\rho$ is a constant slightly greater than 1, and $P$ represents the total persistence value of all items in the data stream $\left(\sum_{e_i \in \mathcal{U}} P_{e_i}\right)$:

$$\Pr\{(P_{e_i} - \hat{P}_{e_i}) \geq \epsilon P\} \leq \frac{1}{2\epsilon P} \cdot (P_{e_i} - \sqrt{P_{e_i}^2 - 4\kappa}), \quad (1)$$

where $\kappa$ is defined as follows:

$$\kappa = Pr_c \cdot Pr_w \cdot \frac{D}{\rho - 1}$$
$$= \left(\left[1 - \left(1 - \frac{1}{w}\right)^{n-1}\right]^d\right) \cdot \left(e^{-\frac{i-1}{w}} \cdot \frac{\left(\frac{i-1}{w}\right)^{d-1}}{(d-1)!}\right) \cdot \frac{M - P_{e_i}}{\rho - 1}. \quad (2)$$

### 4.2 Error Bound of Pontus's Variant

THEOREM 4.2. *Given a small positive number $\epsilon$ and a persistent item $e_i$, the probability that the absolute difference between the actual persistence $P_{e_i}$ and its estimate $\hat{P}_{e_i}$ exceeds $\epsilon P$ is bounded by:*

$$\Pr\{|P_{e_i} - \hat{P}_{e_i}| \geq \epsilon P\} \leq \frac{1}{2\epsilon P} \cdot \left[ (P_{e_i} + E(Y_i)) - \sqrt{P_{e_i}^2 - 4\kappa \cdot \left(1 - \frac{1}{2^\vartheta}\right)} \right] \quad (3)$$

where $P$ is the sum of persistence values of all items, $\kappa$ retains the definition from **Theorem 4.1**, with $\vartheta$ representing the length of a fingerprint. Furthermore, $E(Y_i)$, is given by:

$$E(Y_i) = (M - P_{e_i}) \cdot \frac{1}{2^\vartheta} \cdot \sum_{k=1}^{d} \frac{1}{d} \left[1 - \left(1 - \frac{1}{w}\right)^{n-1}\right]^k. \quad (4)$$

## 5 TRACE-DRIVEN EVALUATION

### 5.1 Setup

To evaluate the performance of Pontus, we conduct evaluations on a laptop equipped with an Intel(R) Core(TM) i5-1135G7 @ 2.40GHz processor and 16GB of DRAM memory, running Ubuntu 20.04 LTS.

**Implementation.** We implement Pontus and the compared benchmarks in C++. For all datasets, we use source-destination addresses as item keys, with each pair consisting of 8 bytes. We employ MurmurHash [5] for hashing incoming items. The number of rows $d$ for Pontus is set to 2 [49], while the number of buckets $w$ in each row is determined based on the specified memory size. For the variant of Pontus, we set the fingerprint length to 18 bits, ensuring a sufficiently low fingerprint collision rate.

**Parameter Settings.** For persistent item lookup, we adjust the window size uniformly across each trace to divide the data into 1,500 windows ($M = 1500$) and set the persistence threshold $\alpha$ to 0.4. In the case of persistent and frequent item lookup, we use a frequency threshold $\beta$ of $5 \times 10^{-4}$ [32]. For persistent and infrequent item lookup, we adjust the frequency threshold $\kappa$ to ensure frequencies remain below 2000. It's important to note that these parameter settings are not fixed; we also explore variations in threshold values to confirm the robustness of Pontus in Section 5.3.

**Baselines.** For persistent item lookup, we evaluate the performance of Pontus against several benchmarks: On-Off Sketch [48], Pyramid-based On-Off Sketch [33], P-Sketch [30], Stable-Sketch [32], Tight-Sketch [31], WavingSketch [29], and Small Space (SS)

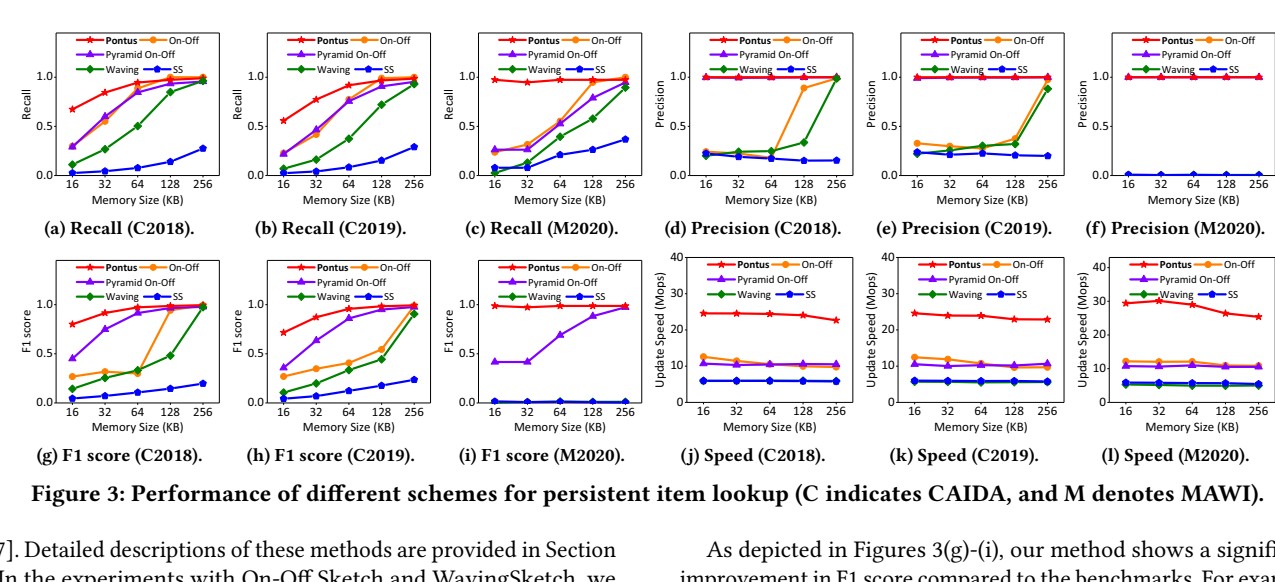

**Figure 3: Performance of different schemes for persistent item lookup (C indicates CAIDA, and M denotes MAWI).**

[27]. Detailed descriptions of these methods are provided in Section 7. In the experiments with On-Off Sketch and WavingSketch, we set the number of slots to 16, following [31]. The parameters for SS are configured according to [47], while for P-Sketch, Stable-Sketch and Tight-Sketch, the number of rows is set to 2 [30].

**Traces.** We select three real-world traces for evaluation: two from the CAIDA 2018 [1] and 2019 [2] datasets, and one from the MAWI 2020 dataset [35]. Specifically, CAIDA 2018 consists of 22.3M packets from 0.76M items, while CAIDA 2019 encompasses 29.5M packets from 1.53M items. MAWI 2020 involves 44.5M packets from 2.75M items.

**Metrics.** We evaluate Pontus's performance on persistence-based tasks using five key metrics: Recall, Precision, F1 score, Average Absolute Error (AAE), and update throughput. Recall measures the fraction of true persistent items correctly identified, while Precision quantifies the proportion of identified persistent items that are truly persistent. The F1 score, calculated as $\frac{2 \times \text{Recall} \times \text{Precision}}{\text{Recall}+\text{Precision}}$, provides a balanced measure of accuracy. For persistence estimation, AAE computes the average magnitude of errors as $\frac{1}{n}\sum_{i=1}^{n}|P_i - \hat{P}_i|$, where $P_i$ is the true persistence and $\hat{P}_i$ is the estimated value. Update throughput measures the processing speed in millions of operations per second (Mops). Each experiment is tested five times, and the average results are reported.

### 5.2 Performance Comparison

*5.2.1 Persistent Item Lookup.* For persistent item lookup, we evaluate Pontus's performance across various memory sizes: 16KB, 32KB, 64KB, 128KB, and 256KB. These memory sizes are consistent with common configurations in recent sketch-based research [22, 32, 46]. *Since P-Sketch, Stable-Sketch, and Tight-Sketch share a similar data structure to Pontus, we compare them separately for clearer analysis.*

Figures 3(a)-(f) show the recall and precision rates of various methods across different datasets. Pontus consistently achieves the highest recall and precision rates under all traces. On average, Pontus improves recall by 18.51%-686.85% for CAIDA 2018, 23.2%-607.29% for CAIDA 2019, and 58.62%-384.21% for MAWI 2020 traces. Additionally, Pontus maintains a precision rate of 1 across all memory budgets. This is because the probability decay mechanism ensures one-sided underestimation errors, meaning that all persistent items identified by Pontus are genuine.

As depicted in Figures 3(g)-(i), our method shows a significant improvement in F1 score compared to the benchmarks. For example, under the CAIDA 2018 trace, the average F1 score of Pontus is 15.11% to 724.17% higher than the baselines. The detection accuracy of our method is even more pronounced under the MAWI 2020 trace. This is because the MAWI trace exhibits higher skewness, indicating a smaller number of persistent items, thereby increasing the detection difficulty. Despite this challenge, Pontus maintains its superiority, resulting in an increase of 45.65% to 10415.54%. Our method excels in its meticulous replacement of persistent items. Given the prevalent highly skewed data distribution in practical scenarios [7, 46], the crude item replacement of On-Off Sketch often leads to many persistent items being mistakenly replaced by non-persistent ones, especially under tight memory constraints where hash collisions are more severe. While WavingSketch uses a Bloom Filter [36] to eliminate duplicates, its application under constrained memory settings results in increased data usage and significant false positive errors. Similarly, in Small-Space, the sampling rate is low under limited memory sizes, leading to reduced lookup accuracy.

Figures 3(j)-(l) illustrate the update speed of different methods. As shown, our method achieves markedly faster update speeds. For example, under the CAIDA 2019 trace, the update speed of our method is 117.31%–320.26% higher than that of the baselines.

**Deep Dive:** We further evaluate the detection accuracy of Pontus against Stable-Sketch, P-Sketch, and Tight-Sketch across different traces and memory settings (16KB and 32KB). As shown in Table 2, Pontus consistently achieves the highest F1 scores compared to the other methods. Pontus's key advantage lies in its efficient use of memory, utilizing only one additional collision decay flag, while the other methods require an additional 2-byte field to track dual-dimensional features. Additionally, Pontus shows faster update speeds, reaching around 25.2Mops, compared to 23.3Mops for P-Sketch, 22.6Mops for Stable-Sketch, and 23.6Mops for Tight-Sketch.

*5.2.2 Persistence Estimation.* Figure 4 shows the error in persistence estimation. Since P-Sketch, Stable-Sketch, and Tight-Sketch do not offer a persistence estimation version, we exclude them from the comparison. As demonstrated, Pontus significantly reduces estimation errors. For AAE, Pontus reduces the average error by 283.56% and 196.16% compared to On-Off Sketch, and by 484.2%

**Table 2: F1 score for Pontus and benchmarks.**

| F1 Score | Pontus | Stable-Sketch | P-Sketch | Tight-Sketch |
|---|---|---|---|---|
| C2018 (16KB) | **0.805** | 0.734 | 0.609 | 0.699 |
| C2018 (32KB) | **0.915** | 0.874 | 0.803 | 0.840 |
| C2019 (16KB) | **0.716** | 0.610 | 0.518 | 0.596 |
| C2019 (32KB) | **0.872** | 0.820 | 0.725 | 0.793 |

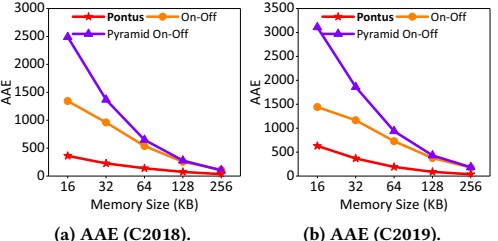

(a) AAE (C2018).          (b) AAE (C2019).

**Figure 4: AAE for persistence estimation.**

**Table 3: Pontus on persistent and (in)frequent item lookup.**

| Memory Size (KB) | 16 | 32 | 64 | 128 | 256 |
|---|---|---|---|---|---|
| F1 score (frequent) | 0.864 | 0.864 | 0.911 | 0.928 | 0.928 |
| F1 score (infrequent) | 0.571 | 0.787 | 0.89 | 0.936 | 0.966 |
| Update speed (Mops, frequent) | 26.076 | 26.472 | 25.4011 | 25.491 | 23.966 |
| Update speed (Mops, infrequent) | 25.464 | 25.247 | 24.108 | 23.143 | 22.644 |

and 396.27% compared to Pyramid On-Off Sketch, over the CAIDA 2018 and 2019 traces, respectively.

Pontus's superiority stems from two key aspects. Firstly, it accurately tracks potential persistent items by monitoring item keys and employing a collision decay flag to protect them from easy eviction by non-persistent ones, leading to precise estimation for items with high persistence. Secondly, Pontus mitigates overestimation errors for non-persistent items by using the smallest value among the hashed buckets. This dual approach allows Pontus to maintain high accuracy across both persistent and non-persistent items.

*5.2.3 Other persistence-based Tasks.* Table 3 presents the F1 score and update speed of Pontus for persistent and (in)frequent item lookup. For persistent and infrequent item detection, we configure $\varsigma$ in the objective function $\Omega$ as 0.2. We employ the CAIDA 2018 trace for testing, and similar trends are observed across other traces. Pontus demonstrates high detection accuracy and rapid update speed. Specifically, with a memory size of 64KB (a typical size of L1 cache [31]), Pontus achieves accuracy scores around 0.9 for both tasks. Furthermore, the update speed exceeds 25Mops, indicating that Pontus is well-suited for high-speed data streams or networks, such as those requiring processing speeds of 14.88Mops per item in a 10Gbps network [42].

When comparing Pontus against the advanced Tight-Sketch [31], focusing on persistent and frequent item lookup using the CAIDA 2018 trace, Pontus outperforms Tight-Sketch in terms of F1 score by 7.46% at 16KB and 3.47% at 32KB. Also, Pontus achieves faster update speeds compared to Tight-Sketch, which is limited to 15.8Mops (figures omitted due to the space limitation).

## 5.3 Multiple Cases

*5.3.1 Performance on Different Parameters.* We use persistent item lookup on the CAIDA 2019 trace as an example, varying the number of windows $M$ to 1000 and 2000, and adjusting the persistence threshold $\alpha$ to 0.3 and 0.5.

**Table 4: Comparative AAE for different memory sizes on CAIDA 2018 and 2019 dataset.**

| AAE | 16KB | 32KB | 64KB |
|---|---|---|---|
| Default Pontus (C2018) | **26.5511** | **11.4344** | **4.0875** |
| Pontus with only $F_{i,j}$ (C2018) | 47.6988 | 17.5055 | 5.074 |
| Default Pontus (C2019) | **39.2512** | **16.4859** | **6.8585** |
| Pontus with only $F_{i,j}$ (C2019) | 67.8754 | 25.8737 | 8.6751 |

Pontus consistently achieves the highest detection accuracy across all parameter settings, demonstrating its effectiveness and robustness (see Appendix B for figures). For example, the average F1 score of Pontus is 25.35%, 20.94%, 20.02%, and 13.62% higher than the closest competing method, Pyramid-based On-Off Sketch, for the (1000, 0.3), (1000, 0.5), (2000, 0.3), and (2000, 0.5) parameter settings, respectively.

*5.3.2 Effectiveness of Utilizing One More Flag.* Pontus utilizes the collision decay flag to improve performance in persistent item lookup. Table 4 presents the AAE of our method with and without the flag. Without the decay flag, other hashed items can prematurely decay the persistence counter before the tracked item arrives in the current time slot. In memory-constrained scenarios, the addition of the collision decay flag significantly reduces estimation errors across different traces. For instance, with 16KB of memory, the flag reduces AAE by 79.65% and 72.93% on the CAIDA 2018 and 2019 traces, respectively.

## 5.4 Memory-Efficient Variant Performance

Due to space constraints, we provide a detailed comparison of the variant and default versions of Pontus in the Appendix C.

## 6 TESTBED EVALUATION

We implement a prototype of Pontus into a real testbed with off-the-shelf Intel Tofino programmable switches using the P4 language. Programmable switches are highly constrained in terms of available resources like number of stages and limited support for mathematical operations. Given these limitations, we implement Pontus with the number of rows $d$ set to 1, consistent with the configurations used in [43, 49].

We employ the Tofino Native Architecture (TNA) RegisterAction extern function to track and update the total persistence value of items, arrival flag $F$, and collision decay flag $R$. We allocate register entries based on the number of buckets used in the experiments. As a hash function to locate items into the registers, we use predefined HashAlgorithm_t.CRC16 function. Mathematical operations like division are challenging in programmable switches due to the constraints of the hardware and the design of the P4 language. The MathUnit extern provides a way to perform approximated division, but it does not yield exact results and is constrained by factors such as approximation methods and input ranges. To overcome this, we calculated better approximated division results offline and fill the table entries in the switch. For the case of probabilistic replacements, we replace the item or decay its persistence counter if the condition $RAND(0, 2^b) \leq int(\frac{2^b}{1+P_{minRow,minIndex}})$ holds, where $b$ is the bit size of the total time windows. Whenever probabilistic replacement conditions hold in the case of collision, we recirculate the packet to replace the non-persistent item or decrease the persistence counter

since TNA does not allow accessing to the same register more than once per packet. In our experiments with CAIDA 2018 trace, only a small fraction (1.93%–2.36%) of the packets are recirculated in different memory allocations due to the probabilistic replacement. At the end of each window, a digest is sent to the controller to reset the associated flags in the register, as the only way of resetting multiple instances in a register array is to use control plane APIs.

**Resource Usage and Latency:** Pontus can detect persistent items effectively at line-rate with a low resource footprint. We evaluate the resource usage and latency of our solution by using Intel P4 Insight analysis tool [24] which provides a detailed analysis of compiled P4 programs. Table 5 shows the resource usage in programmable Intel Tofino Switch. Overall, Pontus consumes 8.2% of the total available switch resources in average which leaves sufficient room in the switch to coexist with other legacy switching functions. Most of the consumption comes from Gateways, used for conditional statements and to implement if/else conditions, and Logical Table IDs that organizes the program into individual lookups to make the processing sequential. The SRAM is the main storage of the switch, used for exact match tables, action data and for stateful objects like registers. Even though we implement our solution in the switch by employing registers, we consume only 4.3% of total available SRAM resources. The results also reveal that Pontus entails an average packet processing latency of 409 ns, incurring just an additional 117 ns of latency compared to simple forwarding in the switch.

**Table 5: Resource Usage of Pontus in Tofino Switch**

| Resource | Usage | Resource | Usage |
|---|---|---|---|
| Hash Bit | 5.7% | Match Crossbars | 4.6% |
| Gateways | 16.7% | Logical Table ID | 21.9% |
| VLIW Instruction | 7.3% | SRAM | 4.3% |
| Total Average | | 8.2% | |

**Accuracy Comparison Between Hardware and Software Versions:** Figure 5 compares the recall rate of capturing real persistent items between the Tofino hardware version and the software version, using the CAIDA 2018 trace. Since our method only produces underestimation errors, all captured persistent items are guaranteed to be real, resulting in a precision of 1, so the precision

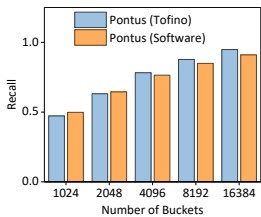

**Figure 5: Recall rate on hardware and software platforms.**

rate is omitted. The results show similar capture rates across various register entry sizes, demonstrating the correctness and consistency of our implementation.

## 7 RELATED WORK

**Persistent Item Lookup:** Conventional methods for persistent item lookup, such as Small-Space (SS) [27], rely on sampling to monitor item persistence while minimizing space usage. However, these methods [10, 27] face significant challenges: they often capture numerous non-persistent items, leading to inefficient memory use. Moreover, reducing sampling rates due to memory limitations can increase lookup errors and compromise accuracy. To overcome

these inefficiencies, researchers have explored various sketch-based approaches. On-Off Sketch [48] periodically increases item persistence but struggles with misidentifying non-persistent items due to its coarse-grained differentiation. Pyramid-based On-Off Sketch [33] enhances counter memory efficiency but inherits the original method's limitations. WavingSketch [29] employs a Bloom Filter to remove duplicates, but this leads to significant false positives and measurement inaccuracies, especially in limited memory scenarios where erroneous replacements occur. PIE [15] employs coding to monitor persistent items, but its complex update process, such as matrix multiplication for item encoding, hampers functionality in memory-constrained environments. P-Sketch [30] and Stable-Sketch [32] use arrival continuity and bucket stability as additional features to better protect potential persistent items. Tight-Sketch [31] incorporates item arrival strength as an extra attribute and employs a two-stage update strategy. However, tracking these features consumes more memory, reducing the number of available buckets and increasing the likelihood of hash collisions.

**Persistence Estimation:** On-Off Sketch [48] estimates item persistence by using a flag to eliminate duplicates within each time window, but this leads to significant overestimation in memory-constrained environments. Pyramid On-Off Sketch [33] improves memory efficiency with a hierarchical counter structure, where overflows are propagated to upper layers. However, this method still faces two challenges under limited memory: (1) slower processing due to complex updates and queries requiring multi-layer traversal, and (2) severe overestimation when multiple counters sharing the same parent overflow simultaneously.

**Persistent and Frequent Item Lookup:** LTC [12] utilizes a complex data structure that incorporates a clock algorithm for increasing item persistence and a long-tail restoration technique to mitigate overestimation errors. However, this sophisticated approach leads to a slow update process, making it unsuitable for high-speed data streams.

**Persistent and Infrequent Item Lookup:** PISketch [18] provides a method for identifying persistent and infrequent items. However, its effectiveness is heavily dependent on complex parameter configurations. This reliance on precise parameter tuning introduces a significant risk of suboptimal performance if the settings are not accurately calibrated to the specific characteristics of the data stream being analyzed.

## 8 CONCLUDING REMARKS

In this paper, we introduced Pontus, a new probabilistic method for accurate and efficient detection of persistent items in high-velocity data streams. Pontus uses two flags to eliminate duplicates within a time window and mitigates underestimation errors. Additionally, we employ a probabilistic eviction strategy to prevent persistent items from being easily displaced by non-persistent ones, addressing the challenge of skewed data distributions. Pontus is further extended to handle other persistence-based tasks, and we provide formal proof of its theoretical soundness. Our extensive trace-driven evaluations demonstrate Pontus's effectiveness, showing significant improvements in lookup accuracy and a substantial increase in update speed. Finally, our implementation on the programmable Tofino switch highlights its feasibility for practical deployment.

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

# A THEORETICAL ANALYSIS

## A.1 Complexity

**THEOREM A.1.** *Assuming that Pontus is configured with $d = \log_2 \frac{1}{\delta}$ and $w = \frac{2}{\epsilon}$, where $\delta$ ($0 < \delta < 1$) represents the error probability and $\epsilon$ ($0 < \epsilon < 1$) denotes the approximation parameter. This parameter indicates that the error in answering a query is within a factor of $\epsilon$ with probability $\delta$. The space usage of our scheme is $O(\frac{1}{\epsilon} \log \frac{1}{\delta} \log n)$. The update time for each operation is $O(\log \frac{1}{\delta} + M \frac{1}{\epsilon} \log \frac{1}{\delta})$. The time complexity of returning all persistent item is $O(\frac{1}{\epsilon} \log \frac{1}{\delta})$.*

**PROOF.** The data structure of our scheme contains $dw$ buckets, where each bucket records a log $n$-bit key, two flag bits, and a persistence counter. Thus, the space usage of our scheme is $O(dw \log n) = O(\frac{1}{\epsilon} \log \frac{1}{\delta} \log n)$.

For our scheme, each operation conducts at most $d$ hash actions to find a proper bucket. Since all counter flags need to be reset to *True* when a new window starts, our scheme will traverse all buckets at the beginning of an epoch. As the number of time windows is $M$, the maximum update time is $O(d + Mdw) = O(\log \frac{1}{\delta} + M \frac{1}{\epsilon} \log \frac{1}{\delta})$.

Returning all persistent items demands a scan of all buckets ($dw$) to obtain which buckets have a persistence counter greater than the predefined threshold, which requires $O(dw) = O(\frac{1}{\epsilon} \log \frac{1}{\delta})$ time. □

## A.2 No Overestimation Errors

**THEOREM A.2.** *For any given item $e_i$, let $P_{e_i}$ and $\hat{P}_{e_i}$ denote the actual persistence and estimated persistence of $e_i$, respectively. Then $\hat{P}_{e_i} \leq P_{e_i}$.*

**PROOF.** At the start of the detection task ($t = 0$), both $\hat{P}_{e_i}$ and $P_{e_i}$ are zero, so the theorem holds. Assume that at the $(t - 1)$-th time window, $\hat{P}_{e_i} \leq P_{e_i}$. At $t$-th time window, two scenarios are possible: (i) If $e_i$ arrives, then $\hat{P}_{e_i} = \hat{P}_{e_i} + 1$ and $P_{e_i} = P_{e_i} + 1$. Hence, $\hat{P}_{e_i} \leq P_{e_i}$ is true. (ii) If an item other than $e_i$ arrives, the estimated persistence of item $e_i$ either decreases by 1 or stays the same, i.e., $\hat{P}_{e_i} = \hat{P}_{e_i} - 1$ or $\hat{P}_{e_i} = \hat{P}_{e_i}$. Given that $P_{e_i} = P_{e_i}$, it follows that $\hat{P}_{e_i} \leq P_{e_i}$.

Since the claim holds for all scenarios, Theorem A.2 is proven. □

## A.3 Error Bound of Pontus

**THEOREM A.3.** *Let $e_i$ denote the item with the $i$-th highest persistence among all considered persistent items. Note that our analysis of the error bound specifically focuses on those persistent items. Given a small positive number $\epsilon$ and a persistent item $e_i$, we define the probability that the difference between $P_{e_i}$ and $\hat{P}_{e_i}$ exceeds $\epsilon P$ by the following inequality. Here, $\rho$ is a constant slightly greater than 1, and $P$ represents the total persistence value of all items in the data stream $\left(\sum_{e_i \in \mathcal{U}} P_{e_i}\right)$:*

$$\Pr\{(P_{e_i} - \hat{P}_{e_i}) \geq \epsilon P\} \leq \frac{1}{2\epsilon P} \cdot (P_{e_i} - \sqrt{P_{e_i}^2 - 4\kappa}), \quad (5)$$

*where $\kappa$ is defined as follows:*

$$\kappa = Pr_c \cdot Pr_w \cdot \frac{D}{\rho - 1}$$

$$= \left(\left[1 - \left(1 - \frac{1}{w}\right)^{n-1}\right]^d\right) \cdot \left(e^{-\frac{i-1}{w}} \cdot \frac{\left(\frac{i-1}{w}\right)^{d-1}}{(d-1)!}\right) \quad (6)$$

$$\cdot \frac{M - P_{e_i}}{\rho - 1}.$$

**PROOF.** We assume that once a persistent item $e_i$ enters a bucket, it is not evicted from the bucket by other items, but its counter may be reduced by other items with a persistence smaller than its own. First, we compute the probability that *any other newly arrived item* $e_j$ does not find an available bucket after being hashed $d$ times, denoted by $Pr_c$:

$$Pr_c = \left[1 - \left(1 - \frac{1}{w}\right)^{n-1}\right]^d, \quad (7)$$

where in a given bucket, the probability that a specific item is mapped is $\frac{1}{w}$. Thus, in a bucket to which $e_j$ is mapped, the probability that no other item is mapped to the same bucket is $\left(1 - \frac{1}{w}\right)^{n-1}$, where $n$ is the total number of distinct items in the data stream. Consequently, in a given array, the probability that a hash collision occurs in the bucket to which $e_j$ is mapped is $1 - \left(1 - \frac{1}{w}\right)^{n-1}$. Thus, $Pr_c$ represents the probability that $e_j$ experiences hash collisions in all $d$ rows.

Next, we define $Pr_w$ as the probability that, during the process of hashing item $e_j$ for $d$ times, $e_i$ remains the item with the lowest persistence count among all rows in which $e_j$ is hashed:

$$Pr_w = \binom{i - 1}{d - 1} \left(\frac{1}{w}\right)^{d-1} \left(1 - \frac{1}{w}\right)^{i-d}. \quad (8)$$

This probability follows a binomial distribution $\text{Bin}(i - 1, \frac{1}{w})$ and can be approximated by a Poisson distribution with parameter $\frac{i-1}{w}$. Therefore, $Pr_w$ can be expressed as:

$$Pr_w = e^{-\frac{i-1}{w}} \frac{\left(\frac{i-1}{w}\right)^{d-1}}{(d-1)!}. \quad (9)$$

Due to the protection of the collision decay flag during the update process in Pontus, reductions in $e_i$'s persistence can only occur during time windows when $e_i$ is absent. Furthermore, within each such period, at most one successful reduction can decrease the persistence count of $e_i$ by 1. Consequently, the maximum decrease $D$ in the persistence count of $e_i$ over the entire process is: $D = M - P_{e_i}$.

Consider the persistence of $e_i$ as consisting of discrete states ranging from 1 to $P_{e_i}$, denoted by $s$, where $s = 1, 2, \ldots, P_{e_i}$. At each state, the probability of successfully reducing $e_i$'s persistence count is $1/s$. Then, based on the previously defined probabilities, the expected number of times the persistence value of $e_i$ is decayed

can be quantified as follows:

$$E(X_i) = Pr_c \cdot Pr_w \cdot D \cdot \frac{1}{E(\hat{P}_{e_i})} \cdot \sum_{g=1}^{E(\hat{P}_{e_i})} \frac{1}{g}$$

$$= \left[ 1 - \left( 1 - \frac{1}{w} \right)^{n-1} \right]^d \cdot e^{-\frac{i-1}{w}} \cdot \frac{\left( \frac{i-1}{w} \right)^{d-1}}{(d-1)!} \qquad (10)$$

$$\cdot (M - P_{e_i}) \cdot \sum_{g=1}^{E(\hat{P}_{e_i})} \frac{1}{g}.$$

The following results apply to our settings where $M$ does not exceed 150K. Items with such high persistence are rare in practice. Consequently, the following formula holds:

$$\sum_{g=1}^{E(\hat{P}_{e_i})} \frac{1}{g} \le \sum_{g=1}^{E(\hat{P}_{e_i})} \rho^{-g} \le \frac{\rho^{-1}(1 - \rho^{-\hat{E}(P_{e_i})})}{1 - \rho^{-1}} < \frac{1}{\rho - 1}, \qquad (11)$$

where $\rho$ is a constant slightly greater than 1, for example, 1.08.

Based on the above, we compute the expected underestimation of $P_{e_i}$:

$$E(\hat{P}_{e_i}) = P_{e_i} - E(X_i)$$

$$\approx P_{e_i} - \frac{Pr_c \cdot Pr_w \cdot D}{E(\hat{P}_{e_i}) \cdot (\rho - 1)} = P_{e_i} - \frac{\kappa}{E(\hat{P}_{e_i})}. \qquad (12)$$

Resolving (12), we obtain $\hat{E}(P_{e_i})$ as:

$$E(\hat{P}_{e_i}) = \frac{P_{e_i} + \sqrt{P_{e_i}^2 - 4\kappa}}{2}, \qquad (13)$$

Furthermore, applying the Markov inequality, we deduce that:

$$Pr\{(P_{e_i} - \hat{P}_{e_i}) \ge \epsilon P\} \le \frac{E(P_{e_i} - \hat{P}_{e_i})}{\epsilon P}$$

$$= \frac{P_{e_i} - E(\hat{P}_{e_i})}{\epsilon P}$$

$$= \frac{1}{\epsilon P} \cdot \left( P_{e_i} - \frac{P_{e_i} + \sqrt{P_{e_i}^2 - 4\kappa}}{2} \right) \qquad (14)$$

$$= \frac{1}{2\epsilon P} \cdot \left( P_{e_i} - \sqrt{P_{e_i}^2 - 4\kappa} \right).$$

Theorem A.3 has been proven. □

To validate the correctness of the derived error bound, we utilize the CAIDA 2018 trace for testing, which comprises 2.3M packets from 0.76M items. Initially, we compare the real and estimated values of persistent items using our method with a memory size of 32KB, adhering to the settings described in Section 5.1. Specifically, Pontus captures 883 persistent items. For clarity, in Figure 6(a), we randomly select 200 items from this pool, noting that all other items yielded similar results. As depicted, our estimated persistence closely aligns with the actual persistence, with no instances of overestimation errors.

Then, we validate the accuracy of our derived bound. We set $\rho$ at 1.08 and vary the memory size from 32KB to 256KB to cover typical L1 cache sizes, specifically 32KB and 64KB. Additionally, we adjust the parameter $\epsilon$ to maintain $\epsilon P$ at 20, signifying that the persistence

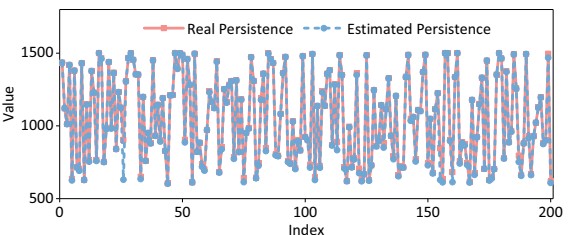

(a) Comparison between actual and estimated persistence by Pontus.

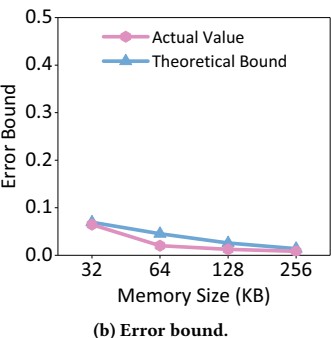

(b) Error bound.

Figure 6: Model Validation.

estimation errors are within 20. Our observations in Figure 6(b) indicate that the theoretical error bound closely matches and is slightly larger than the experimental values, thus confirming the correctness of our derived error bound.

## A.4 Error Bound of Pontus's Variant

THEOREM A.4. *Given a small positive number $\epsilon$ and a persistent item $e_i$, the probability that the absolute difference between the actual persistence $P_{e_i}$ and its estimate $\hat{P}_{e_i}$ exceeds $\epsilon P$ is bounded by:*

$$Pr\{|P_{e_i} - \hat{P}_{e_i}| \ge \epsilon P\} \le \frac{1}{2\epsilon P} \cdot \Bigg[ (P_{e_i} + E(Y_i))$$

$$- \sqrt{P_{e_i}^2 - 4\kappa \cdot \left( 1 - \frac{1}{2^\vartheta} \right)} \Bigg] \qquad (15)$$

*where $P$ is the sum of persistence values of all items, $\kappa$ retains the definition from **Theorem A.3**, with $\vartheta$ representing the length of a fingerprint. Furthermore, $E(Y_i)$, is given by:*

$$E(Y_i) = (M - P_{e_i}) \cdot \frac{1}{2^\vartheta} \cdot \sum_{k=1}^d \frac{1}{d} \left[ 1 - \left( 1 - \frac{1}{w} \right)^{n-1} \right]^k. \qquad (16)$$

PROOF. First, by using fingerprints instead of keys, for any new incoming item $e_j$ to successfully reduce the probability of $e_i$, it is required that the fingerprints of the two items differ. The probability of this occurring is:

$$Pr_{fp} = 1 - \frac{1}{2^\vartheta}. \qquad (17)$$

We define $E(Y_i)$ to represent the expected overestimation in the persistence count of $e_i$ due to fingerprint collisions:

$$E(Y_i) = (M - P_{e_i}) \cdot \frac{1}{2^{\vartheta}} \cdot \sum_{k=1}^{d} \frac{1}{d} \left[ 1 - \left( 1 - \frac{1}{w} \right)^{n-1} \right]^k, \quad (18)$$

where the persistent count of $e_i$ can only be increased by at most 1 in each period in which it is not present. The factor $1/2^{\vartheta}$ represents the probability that two items share the same fingerprint. The last term represents the probability that any newly arriving $e_j$ hashes into the same bucket as $e_i$ during a given time window by sharing the same fingerprint.

Based on these definitions, the expected value of $\hat{P}_{e_i}$ under the fingerprint scenario is calculated as follows:

$$\begin{aligned} E(\hat{P}_{e_i}) &= P_{e_i} - E(X_i) + E(Y_i) \\ &\approx P_{e_i} - \frac{Pr_c \cdot Pr_w \cdot D}{E(\hat{P}_{e_i}) \cdot (\rho - 1)} \cdot \left( 1 - \frac{1}{2^{\vartheta}} \right) + E(Y_i) \\ &= P_{e_i} - \frac{\kappa}{E(\hat{P}_{e_i})} \cdot \left( 1 - \frac{1}{2^{\vartheta}} \right) + E(Y_i), \end{aligned} \quad (19)$$

where $Pr_c$, $Pr_w$, and $D$ maintain their definitions as outlined in **Theorem A.3**.

Similar to the above derivation process, we solve formula (19) to obtain:

$$E(\hat{P}_{e_i}) = \frac{\left[ P_{e_i} + E(Y_i) \right] + \sqrt{P_{e_i}^2 - 4\kappa \cdot \left( 1 - \frac{1}{w^{\vartheta}} \right)}}{2}, \quad (20)$$

Therefore, by applying the Markov inequality, we deduce the error bound as:

$$\begin{aligned} Pr \left\{ |P_{e_i} - \hat{P}_{e_i}| \geq \epsilon P \right\} &\leq \frac{E(P_{e_i} - \hat{P}_{e_i})}{\epsilon P} \\ &= \frac{P_{e_i} - E(\hat{P}_{e_i})}{\epsilon P} \\ &= \frac{1}{2\epsilon P} \cdot \left[ \left( P_{e_i} + E(Y_i) \right) \right. \\ &\quad \left. - \sqrt{P_{e_i}^2 - 4\kappa \cdot \left( 1 - \frac{1}{2^{\vartheta}} \right)} \right]. \end{aligned} \quad (21)$$

$\square$

# B PERFORMANCE ON DIFFERENT PARAMETERS

Here, we use persistent item lookup as an example. We vary the number of windows, $M$, to 1000 and 2000, and adjust the persistence threshold, $\alpha$, to 0.3 and 0.5. We employ the CAIDA 2019 trace for testing.

As shown in Figure 7, we observe that Pontus maintains the highest detection accuracy across different parameter settings, affirming its effectiveness and robustness. For instance, the average F1 score of Pontus is 25.35%, 20.94%, 20.02%, and 13.62% higher than the most competing method Pyramid-based On-Off Sketch over (1000, 0.3), (1000, 0.5), (2000, 0.3), and (2000, 0.5) parameter settings.

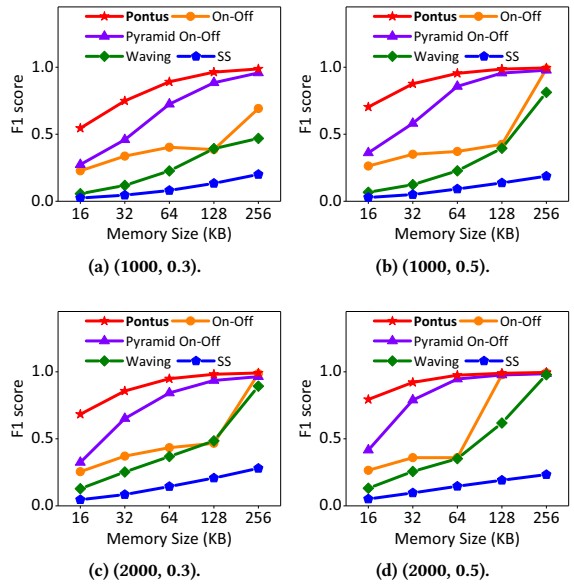

**(a) (1000, 0.3).**     **(b) (1000, 0.5).**

**(c) (2000, 0.3).**     **(d) (2000, 0.5).**

**Figure 7: F1 scores of different methods under various parameter settings for persistent item lookup (where the first value indicates $M$ and the second value indicates $\alpha$).**

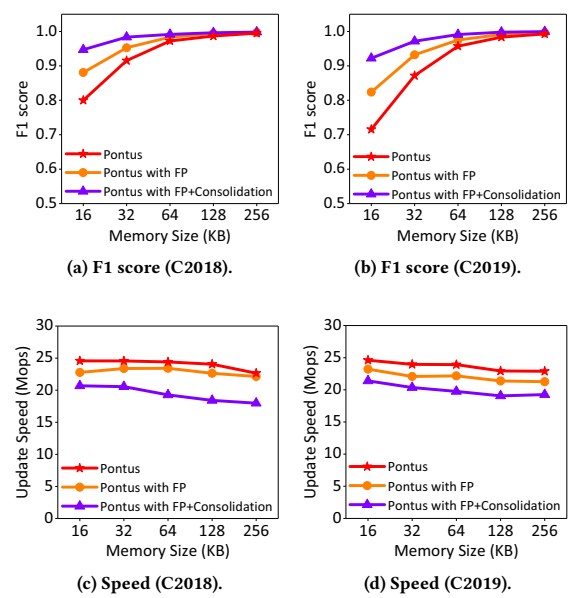

**(a) F1 score (C2018).**     **(b) F1 score (C2019).**

**(c) Speed (C2018).**     **(d) Speed (C2019).**

**Figure 8: F1 score and update speed of our default Pontus version and its variant for persistent item lookup.**

# C VARIANT PERFORMANCE

Figure 8 depicts the results of our default Pontus and its variant across various traces. Our variant incorporates two techniques: fingerprint (PF) compression and field consolidation. We evaluate the effectiveness of each component. As depicted in Figure 8(a) and

(b), each technique demonstrates improvements in detection accuracy, particularly under tight memory constraints such as 16KB. For example, with the CAIDA 2018 trace and a memory size of 16KB, our variant enhances accuracy by 18.39% compared to our default version. On average, the variant improves accuracy by 5.3%. However, such improvements come at the price of lower processing speed. As shown in Figure 8(c) and (d), we observe that the use of fingerprint slows down the update speed. This is because each item requires additional hash operations to obtain its fingerprint value.

Furthermore, counter consolidation further reduces the speed, as it involves checking whether the consolidation is successful and operating on a bit-level, which leads to slower processing. Additionally, the use of fingerprint breaks invertibility. During the query process, each item needs to be rehashed to obtain its fingerprint, resulting in longer query times. Thus, if speed is not a primary concern in practice, using this variant can be an option. By default, we choose the default version because it achieves superior accuracy compared to existing baselines and maintains high processing speed.

