# OpenReview forum: "Pontus: A Memory-Efficient and High-Accuracy Approach for Persistence-Based Item Lookup in High-Velocity Data Streams"
_ACM.org/TheWebConf/2025/Conference — WWW 2025 Oral_

### Official Review · Reviewer_i545 · 2024-12-01

**Novelty:** 5
**Technical Quality:** 6

**Review:**

This paper focuses on addressing the challenges of Persistent Item Lookup and Persistence Estimation problems. Pontus introduces an innovative persistence tracking strategy by leveraging a flag-based design (arrival flag $F$ and collision decay flag $R$), which ensures precise tracking of item persistence and minimizes the impact of hash collisions while maintaining memory efficiency. Experimental results on multiple datasets demonstrate the effectiveness of the proposed method under different scenarios.

Quality: This work is technically sound.

Clarity: The manuscript is well-organized and clearly written.

Originality: This work contributes new insights to the field.

Significance: This work offers incremental contributions that enhance existing research and practice in this area.

Pros:

S1: Comprehensive Experiments: The paper thoroughly tests the algorithm on various datasets, proving its effectiveness.

S2: Versatility: The method works well for different types of persistent item detection tasks.

Cons:

W1: In Algorithm 1, if my understanding is correct, the initialization of the P and K should only occur once for all windows, whereas the initialization of the F and R flags should be updated at the beginning of each window. The current algorithm flow appears somewhat confusing on this point.

W2: In the experiments, the number of rows $d$ for Pontus is set to 2, and the fingerprint length is set to 18 bits. These parameters seem to have a significant impact on the algorithm's performance. Providing an experimental analysis of their influence and the rationale behind choosing these specific values would be highly valuable.

W3: For the evaluation of Persistent and Frequent Item Lookup and Persistent and Infrequent Item Lookup scenarios, the paper only demonstrates the performance of Pontus. Including LTC and PISketch mentioned in related work as baselines would provide a more comprehensive comparison.

W4: The paper needs to discuss the potential application scenarios in the web field and the possible limitations of the proposed algorithms.

**Questions:**

W1-W4

**Reviewer Confidence:**

4: The reviewer is certain that the evaluation is correct and very familiar with the relevant literature

**Scope:**

3: The work is somewhat relevant to the Web and to the track, and is of narrow interest to a sub-community

---

### Official Review · Reviewer_wEcF · 2024-12-01

**Novelty:** 5
**Technical Quality:** 5

**Review:**

This paper presents Pontus, a sketch-based system designed for fast, memory-efficient, and accurate persistent item detection.
Persistent item detection and its derived applications, such as frequent/infrequent item detection, encounter challenges including accuracy and memory efficiency.
Potnus proposes a sketch-based detection mechanism by (1) deploying new flag bits to avoid errors by hash collision, (2) implementing a probability-decay eviction strategy to maintain the persistent items, and (3) compressing the key and value field to improve memory efficiency. Evaluations demonstrate that Potnus present high effectiveness among different traces compared to baselines. Implementation on Tofino also proves it can be migrated to many hardware targets.
This paper is easy to follow, and the logic is fluent. It focuses on a basic function component on current web security or management applications, persistent item detection showing considerable significance. This work still presents its novelty via optimizing the sketch data structure and updating mechanism, although employing sketches into persistent item detection has been studied by many existing studies.

Pros:

* This paper is easy to follow with good writing.
* Optimizing sketch data structure and updating mechanism for high effectiveness.
* Extensive evaluations on different metrics, baselines, and datasets.

Cons:

* The evalatuions can be refined with more details about real-world web applications.
* The real testbed evalatuions are underspecified.

**Questions:**

I have some questions on evaluations.

* The evaluations mainly focus on abstract persistent frequent/infrequent items. However, since it plays an important role in web security applications, as mentioned in the introduction, what will the performance improvement or overhead be if it is integrated into current typical applications? I think additional evaluations or analyses can help to explain its effectiveness.
* Implementation on the real testbed can be discussed more in detail. For example, the window size (period or packet number?) on P4 version should be specified. Since the register reset at the end of each window is carried out by the controller, whose latency is about 10 ms to 100 ms, will this induce a race condition, or is the window size much larger so that it is negligible?

**Reviewer Confidence:**

3: The reviewer is confident but not certain that the evaluation is correct

**Scope:**

3: The work is somewhat relevant to the Web and to the track, and is of narrow interest to a sub-community

---

### Official Review · Reviewer_kjSJ · 2024-12-02

**Novelty:** 6
**Technical Quality:** 7

**Review:**

This is a well-motivated, prepared and communicated study to present what authors call Pontus, an approach to track persistent items in web streams. Authors present the approach and convincing evaluation results against the existing approaches. In my view this paper is publishable as it is, and should provide an interesting contribution for the conference.

**Questions:**

Are you considering to release the implementatation as open source? I am sure there would be plenty of interest in it for research and application purposes.

I wonder whether you used LLMs in any way as part of your study, or for preparing the submitted paper? If so, I recommend you to clearly state the role of LLMs & how they helped you.

**Reviewer Confidence:**

4: The reviewer is certain that the evaluation is correct and very familiar with the relevant literature

**Scope:**

3: The work is somewhat relevant to the Web and to the track, and is of narrow interest to a sub-community

---

### Official Review · Reviewer_zRkM · 2024-12-03

**Novelty:** 4
**Technical Quality:** 4

**Review:**

**Summary**

This paper introduces a novel method Pontus, designed to efficiently and accurately detect persistent items in high-speed data streams. Experiments demonstrate its effectiveness in improving detection accuracy and processing speed.


**Strengths**

(1) The studied problem is meaningful for the enhancement and development of network security and system performance optimization.

(2) A theoretical feasibility analysis of the proposed method is included in the paper.

(3) Extensive experimental validation is provided to prove the effectiveness and practicality of the proposed approach.

**Weaknesses**

(1) The proposed method appears to be a pipeline design, using various existing technical approaches to filter and count persistent items, while improving memory utilization efficiency through methods such as Fingerprint-based counter merging.

(2) Is the proposed approximate data structure suitable for most network applications? Would using this data structure require replacing existing data management patterns, potentially leading to incompatibility or deployment difficulties in many applications?

**Questions:**

Please refer to the Weaknesses.

**Reviewer Confidence:**

1: The reviewer's evaluation is an educated guess

**Scope:**

3: The work is somewhat relevant to the Web and to the track, and is of narrow interest to a sub-community

---

### Official Review · Reviewer_KFqr · 2024-12-05

**Novelty:** 6
**Technical Quality:** 7

**Review:**

The authors of this paper propose an approach that uses sketches designed for the detection of persistent items from a data stream.

The paper is well-motivated in the introduction section. The related work is well presented, and the most important papers in this area are well cited.

The design of Pontus includes two flags: one arrival flag "F" to track whether the item arrived during the current window or before it, and a second collision decay flag "R" to indicate if a hash collision occurred in the bucket during the current window with the main goal of avoiding further decrease in the decay factor.

Similar to other stream sketches, Pontus includes two main operations: update and query operations. A running example is provided to illustrate all parts of the proposed update algorithm. An extended example would be more useful to illustrate the main loop of the algorithm and the impacts of Case 3 probabilistic replacements.

Pontus can be used to lookup frequent and infrequent items. Additionally, authors propose memory-efficient variations of Pontus by using fingerprint-based key compression and field consolidations.

A theoretical analysis of error bounds is provided, with details included in the appendix of the paper. An experimental evaluation compares the proposed solution to state-of-the-art algorithms. A separate implementation and experiments are provided using Intel Tofino programmable switches using the P4 language.

**Questions:**

1. How important is the probabilistic replacement? How would the results and error ratio be changed if "case 3" from the algorithm were removed? How would it change the error ratios?

2. Are the performance results shown in Figure 3 within the expected theoretical error upper bounds?

**Reviewer Confidence:**

3: The reviewer is confident but not certain that the evaluation is correct

**Scope:**

4: The work is relevant to the Web and to the track, and is of broad interest to the community